# “I Have Nowhere to Go”: A Multiple-Case Study of Transgender and Gender Diverse Youth, Their Families, and Healthcare Experiences

**DOI:** 10.3390/ijerph18179219

**Published:** 2021-09-01

**Authors:** Megan S. Paceley, Jennifer Ananda, Margaret M. C. Thomas, Isaac Sanders, Delaney Hiegert, Taylor Davis Monley

**Affiliations:** 1School of Social Welfare, University of Kansas, 1545 Lilac Lane, Lawrence, KS 66045, USA; jananda@ku.edu (J.A.); trdavis0518@gmail.com (T.D.M.); 2Luskin School of Public Affairs, University of California Los Angeles, 3250 Public Affairs Building, Los Angeles, CA 90095, USA; thomas@luskin.ucla.edu; 3A Way Home Washington, 1200 12th Ave. S. Suite 710, Seattle, WA 98144, USA; isanders@awayhomewa.org; 4School of Law, University of Kansas, 1535 W 15th Street, Lawrence, KS 66045, USA; dchiegert@gmail.com

**Keywords:** transgender, healthcare, children, adolescents, Midwest

## Abstract

Transgender and gender diverse (TGD) youth experience health disparities due to stigma and victimization. Gender-affirming healthcare mitigates these challenges; yet, we have limited understanding of TGD youth’s healthcare experiences in the U.S. Midwest and South. Using a multiple case study design, we aimed to develop an in-depth and cross-contextual understanding of TGD youth healthcare experiences in one Midwestern state. Families with a TGD child under 18 were recruited with the goal of cross-case diversity by child age, gender, race, and/or region of the state; we obtained diversity in child age and region only. Four white families with TGD boys or non-binary youth (4–16) in rural, suburban, and small towns participated in interviews and observations for one year; public data were collected from each family’s community. Thematic analysis was used within and across cases to develop both family-level understanding and identify themes across families. Findings include a summary of each family as it relates to their child’s TGD healthcare experiences as well as the themes identified across cases: accessibility and affirming care. Although limited by a small sample with lack of gender and race diversity, this study contributes to our understanding of TGD youth healthcare in understudied regions.

## 1. Introduction

Transgender and gender diverse (TGD) youth experience stigma and victimization in their homes, schools, and communities [1]; these experiences are associated with increased health concerns, including depression, suicidality, substance abuse, eating disorders, and stress [2,3,4,5]. These health concerns can be mitigated or alleviated by supportive individuals and communities. When TGD youth report having at least one supportive adult in their lives, their risk of suicidality decreases by 40% [6]. Additionally, TGD youth report decreased depression and suicidality when people in their lives use their chosen names [7]. Communities can also reduce risk for TGD youth via TGD-inclusive non-discrimination policies, presence and visibility of other TGD people, and access to affirming resources [8].

An important resource for TGD youth is access to physical and mental healthcare that is affirming of their gender identities and expressions [8]. At the broadest level, having affirming healthcare means access to providers who use a patient’s chosen name and pronouns and affirms their gender. More specifically, gender-affirming healthcare for youth includes access to puberty blockers (to safely delay puberty associated with sex assigned at birth), use of hormones consistent with their gender, and medications to promote physical development [9]. Gender-affirming healthcare is recognized by major pediatric medical organizations as evidence-based practice for TGD youth [10,11] and is associated with reduced health disparities [12] including risk of suicide [13].

Even with this professional recognition, gender-affirming care is facing significant social and political backlash. In early 2021, in the U.S. more than 20 bills were proposed that aim to restrict gender-affirming healthcare for TGD children and adolescents [9,14]. These bills include provisions making it a criminal act to provide gender-affirming care (for medical professionals and/or parents), opening medical professionals up to civil liabilities for providing gender-affirming care to youth, and/or restricting insurance coverage for gender-affirming care for people under 18 [9]. If passed, these bans could restrict access to gender-affirming care for as many as 45,000 TGD youth in the U.S. [9], primarily in Midwestern and Southern states.

Access to affirming physical and mental healthcare is a critical component of reducing health disparities for TGD youth, yet we lack research on the healthcare experiences of TGD youth in the U.S. Midwest and South, precisely those areas where TGD youth face heightened risks for health disparities. Simultaneously, legislation in these regions is targeting TGD youth via restrictive and discriminatory policies and rhetoric. To redress this critical gap in the literature, this study centers the lived experiences of TGD youth and their families in one Midwestern state with the goal of understanding their healthcare experiences broadly and related to gender-affirming care specifically.

### 1.1. TGD Youth in the Midwest

The Midwest is characterized as having low tolerance for sexual minorities [15]. TGD youth in the Midwest and South report higher rates of anti-TGD victimization compared with TGD youth in other regions of the country [16,17]. Few studies have centered an understanding of the Midwest context on the experiences and well-being of TGD and sexual minority youth. One study explored the community context of TGD youth living in rural communities in a Midwestern state [18]. They found that community climate (e.g., the level of support for TGD people in a community) was a better predictor of supportive resources than was community size. This finding suggests that it is critical to identify strategies to increase support for TGD people within their communities. Another study examined the community factors relevant to how TGD youth perceive support in rural and small towns in the Midwest [8]. Youth described a lack of TGD-affirming resources, including healthcare options that affirmed and supported their identities.

### 1.2. TGD Youth Healthcare

Access to supportive and competent healthcare can reduce health concerns and promote well-being for TGD youth [19], yet TGD youth face barriers to accessing both general and gender-affirming healthcare. Specifically, barriers to healthcare for TGD youth include fears of mistreatment based on gender [20,21] and providers using youth’s incorrect names/pronouns [20,22]. Additionally, TGD youth face barriers to gender-affirming care such as lack of competent providers [21,22,23], lack of access to puberty blockers or hormones [20,22], and insurance and financial issues [21,22]. Alternatively, affirmation of TGD youth’s gender is associated with reduced barriers to affirming care for TGD youth of color [24,25].

Importantly, access to supportive and affirming healthcare extends beyond provider competency and accessibility. For TGD youth under the age of 18, parental consent and support is often a prerequisite to care [26]. Very few studies have examined the healthcare experiences of TGD youth from the perspective of both youth and parents. A qualitative study of TGD youth and parents in Newfoundland found that while both youth and parents identified the barriers described above, parents also discussed concerns about their child’s mental health, wait times for access to care, and a lack of information on providers and gender-affirming care [23]. Additionally, youth specifically worried about parental acceptance, safety, and feelings of dysphoria.

These studies provide an important understanding of the healthcare experiences of TGD youth, and yet there remain key gaps in the literature. Given the important role of parents in the provision of healthcare for TGD youth, we need more studies including both youth and parent perspectives. Additionally, healthcare access and affirmation may be different in regions and communities with less supportive attitudes and resources, such as the U.S. Midwest and South. Finally, existing studies have primarily used cross-sectional surveys or interviews, providing important details at a point in time. We lack studies exploring the lived experiences of TGD youth and their families over time. Therefore, the purpose of this study was to center the lived experiences of TGD youth and their families in one Midwestern state over the course of a year with the goal of learning about their experiences with healthcare.

## 2. Materials and Methods

We used a multiple case study design to explore the healthcare experiences of TGD youth and their families in one Midwestern state. Multiple case study designs allow for in-depth exploration of a phenomenon across varying contexts or situations [27]. Four families in one Midwestern state participated in the study over the course of twelve months (July 2019–June 2020). All procedures were approved by the University of Kansas ethics review board.

### 2.1. Positionality

As scholars engaged in research-advocacy with TGD youth, it is essential that we identify our positionalities within the study. Author one (she/they) is a queer, white, genderqueer woman with professional, educational, and class privilege. They have over a decade of practice experience working with TGD youth. They led the study, engaged in all data collection and analysis, and were the primary contact for all families. Author two (she/her) is a white, queer, cisgender woman with professional, educational, and class privilege. She engaged in data analysis and dissemination. Author three (she/her) is a white, lesbian, cisgender woman, with class, educational, and professional privilege. She joined the study during the dissemination phase as a policy expert. Author four (they/them) is a Black, two spirit, queer person with professional and educational privilege. They joined the study during the dissemination phase as a TGD practice expert. Author five (they/them) is a white, queer, transmasculine nonbinary person with professional, educational, and class privilege. They engaged in data analysis and dissemination. Author six (she/her) is a white, queer, cisgender woman. She was involved in the study from the start and engaged in data collection, analysis, and dissemination.

### 2.2. Recruitment and Sampling

A multiple case study design requires the selection of cases (i.e., families) who have a similar connection to a phenomenon (i.e., TGD healthcare) but with different contexts or situations [27]. We aimed to recruit and retain four families who lived in the sampled state and had at least one child under the age of 18 who identified as transgender, non-binary, or gender diverse. We also aimed to include varying contexts to explore diversity across cases; specifically, we hoped to include children of varying age, race/ethnicity, and gender identities or expressions, as well as families in multiple regions of the state.

Families were recruited via advertisements on social media, emails from LGBTQ+ organizations, and professional connections throughout the state. Interested parents or children were asked to email or call the first author to establish eligibility and consent. For each family, the parent was the initial contact. A total of five families consented to participate. One family withdrew after data collection began, and their data are not included. Although we attempted to recruit a diverse sample in terms of race/ethnicity and gender, the final set of families included only age and regional diversity. Table 1 describes the characteristics of each family. Notably, all families were white and three youth participants were trans boys.

### 2.3. Data Collection

Data collection included three or four in-depth interviews with parent and child participants, observations of each family, public document review, and member checking/data review by each family. Data collection occurred between July 2019 and June 2020. Due to the onset of COVID-19 in early 2020, all data collection after March 2020 was conducted online.

#### 2.3.1. Interviews

Multiple interviews were conducted with each family to understand their individual contexts and experiences, attending to changes over time. After receiving initial agreement to participate in the study by parents and children, we set up a time to meet the families in person. The first and last author traveled to meet the families in their homes. At the first meeting, we brought food, snacks, and/or things to play with (for the family with younger children) to establish rapport and get to know the family before starting data collection. Consent documents were shared and parents provided signed consent for themselves and their child. Children received an assent form and provided verbal agreement to participate. Families participated in three to four interviews over the course of twelve months, spaced about three to four months apart. Parents and children received a USD 20 gift card for each data collection visit for a total of up to USD 160 per family.

An interview guide was used to facilitate each interview. The first interview guide was developed to establish an understanding of the child, family, school, community, and healthcare contexts. Initial interviews with parents and children were conducted separately to provide privacy and confidentiality for all families except one (due to age of the child). The parent interview guide included questions related to their interest in the study, family background, their TGD child, their child’s physical and mental healthcare experiences, and needs of the family. The child interview guide was adapted based on the age of the child and included questions related to their family, gender, healthcare experiences, and any need for support they had. Subsequent interview guides were developed for each family based on their individual contexts and after preliminary analyses of their previous interview(s). Subsequent interviews focused on changes since previous meetings, past and upcoming healthcare appointments, shifts related to gender identity or expression, and ongoing or new needs for support. Two families elected to conduct these subsequent interviews as a family, rather than individually (both were the two families with younger children). All interviews were audio recorded and transcribed by a professional transcription service.

#### 2.3.2. Observations

Participant observations were used to document and explore family dynamics and contexts. At each family meeting, we engaged with the parent(s) and child together before and after interviews. Following each meeting, we individually documented our observations of the family, family dynamics, and possible topics for further exploration. Observations included memos from the interviews. Observation memos were written within 48 h of each meeting, and we also discussed our reactions and thoughts as a group.

#### 2.3.3. Public Document Review

Public documents in each family’s community were collected to understand the healthcare and community context for each family. Public data included the presence or absence of TGD-affirming healthcare services, non-discrimination policies inclusive of gender identity, and any other relevant public documents, such as lists of TGD-affirming healthcare providers.

#### 2.3.4. Member Checking/Data Review

At the end of the data collection period (Summer, 2020), each family was provided with a two-to-three-page summary description of their family contexts and TGD healthcare experiences. They were asked to read and review it for accuracy and anonymity, allowing them the opportunity to change details or obscure facts that might make them easy to identify. No families asked to change any information. They were asked to answer a few questions via email in their response, including whether there had been any changes or updates since their last interview, feedback they hoped healthcare providers would take from this study, and ideas for training healthcare professionals. All families responded to the questions, and these responses were incorporated into the analyses.

### 2.4. Data Analysis

Data for this study included interview transcripts, observation notes, public documents, and member checking reviews. Given the multiple case study design, it was essential to analyze these documents within cases (e.g., each family) and across cases (e.g., the entire data set). This allowed for an in-depth understanding of each family’s contexts and experiences and for the comparison of data across cases. This is an essential step in a multiple case study with the goal of understanding each individual case and identifying similarities and differences across cases [27].

The first and last authors began analyzing data within families at the start of data collection to develop a thematic understanding of each family prior to the subsequent interview/meeting. Interview transcripts and observation notes were read by each author who each took notes about emerging ideas and codes to attend to in subsequent rounds of analysis. These notes were used to create interview guides for subsequent meetings with families, including follow-up questions from within-case analysis (e.g., “Tell me more about what happened when the doctor….”) and across cases (e.g., “Some families are reporting [a specific experience]. Has that been true for you).

When data collection ended, the first and final two authors engaged in the thematic analysis procedures outlined by Braun and Clark [28]. Each author read all data materials to establish familiarity within each family and then across cases. Next, codes were generated within data related to healthcare experiences, needs, and concerns. We developed themes and then reviewed them to ensure accuracy and that no other codes or themes were missing. One author then wrote family summaries using these themes and sent them to the families for review. Families were able to make corrections or omissions and answered a few final questions; these answers were used in the final round of analyses. Themes were then compared across cases by coding excerpts for similarities and differences and compiling them into a set of themes that described the findings across all four cases.

## 3. Results

With the goal of providing a rich description of each case, as well as sharing insights across cases, the findings from this study are presented in two parts. First, we provide a brief but in-depth description of each family and their healthcare experiences. Second, we share the findings from the across-case analyses demonstrating two broad themes of TGD healthcare experiences: accessibility and affirming care.

### 3.1. Family Descriptions

#### 3.1.1. Family 1

Travis was 16 years old at the start of the study, the middle child of three siblings, living with his mom and dad in a rural town. Although adjacent to an urban county, the community had a small town feel with neighbors waving at cars as they drove by and a small, walkable downtown in which we picked up lunch for the family on our first visit. There were no TGD-affirming providers within the county, and the county had no legal protections for TGD people. An urban city is about an hour away, providing access to resources, but only if one were able to make the drive. Throughout the course of the study, Travis’ family had access to medical care and private insurance. Travis and his mom, Carrie, were open, loving toward each other, and friendly. Travis shared his love of music, art, and writing.

Travis explored his gender on his own for years prior to telling his parents he is trans at the age of 13. They were supportive of him, using his correct name and pronouns, finding TGD-affirming providers, and advocating for him. Travis indicated his school was mostly supportive, but he described instances where, in an attempt to be supportive, the school would do things like ask what group he preferred when events were separated by gender, which he felt like they should know, given he is a boy. Although he was generally reserved, he had gotten involved in advocacy efforts with his school’s gender and sexuality alliance (GSA). Travis attended an LGBTQ+ organization in the neighboring city, providing him with social support and resources.

Travis sought both TGD-related medical and mental healthcare. He described his therapist as helpful and affirming. He first saw his family doctor for gender-related care; however, the doctor was uncomfortable with this and provided no referrals. Carrie explored other options and eventually found a transgender-friendly clinic, referred by friends from the LGBTQ+ center. Travis receives hormones and ongoing care from the clinic, which has been affirming and competent; physicians and nurses use proper language, understand how to work with TGD youth, and ensure TGD people feel comfortable. However, the clinic is an hour away, and there have been logistical challenges with scheduling and physician turnover. Carrie described several examples of times in which their insurance initially would not cover visits to the clinic due to coding issues or confusion about Travis’ gender. After phone calls to the insurance, these issues were resolved; however, it required time and emotional labor on Carrie’s behalf. She described their healthcare hurdles to be finding and accessing TGD-affirming healthcare and navigating insurance challenges, such as improperly coding TGD-related healthcare as not covered.

#### 3.1.2. Family 2

Aiden was 15 years old, the youngest of two, living with his mom and dad in a suburban community. The town had a long-standing non-discrimination ordinance that included gender identity, as well as numerous therapists skilled at working with TGD youth. There were a few medical providers affirming of TGD people; however, there were no TGD-specific providers or centers. The city is about an hour from an urban community that had numerous TGD-affirming providers, including a gender clinic for children, if families could make the drive. Aiden and his family had lived in this community for a couple of years, having moved from another state due to transphobia. They had access to private insurance and means to travel and pay for medical care. Aiden shared enjoying physical activity and playing video games. Throughout the study, Aiden and his mom, Vanessa, as well as other family members who stopped in to say hi, were friendly and welcoming. Aiden and Vanessa appeared to have a strong, trusting relationship.

Aiden knew at five that his gender was not what people expected of him, but he did not have the words for it at such a young age. Vanessa described wondering if her child might be transgender and meeting with a mental health professional to ensure they were providing Aiden with the right types of support. At 10, he met with a gender-affirming therapist and then came out as transgender, changing his name and pronouns and starting hormone blockers to delay puberty. His family has been incredibly supportive, advocating for him when his school denied him use of facilities consistent with his gender (e.g., boys’ bathrooms), and eventually moving to provide him with a more accepting community and resources.

Aiden’s experiences with healthcare have been mixed. He described his previous doctor as “scared” of him, as he refused to prescribe hormones even after a therapist said Aiden was ready. He switched doctors and found a general practitioner who was TGD-affirming. He also visited the gender clinic in the neighboring urban city every four weeks. He sees a local therapist who is skilled at working with TGD youth. Aiden and Vanessa described generally positive experiences with healthcare but indicated that barriers existed with physician training and education on how to work with TGD people.

#### 3.1.3. Family 3

Blake was 11 years old, the middle child of three kids living with their mom, Faith; Blake’s dad lived in another state. Although they lived within an urban city, their home was situated on the outskirts of town, just off a major interstate, giving it a more rural feel. Despite that, the city is one of the largest in the state and has several LGBTQ+ organizations and TGD-affirming healthcare providers. At the time of this study, the city did not have any policies protecting TGD people from discrimination. Blake shared his joy of cats, art, and rock collecting, as well as experimenting with make-up and fancy clothes. During our meetings, the family often had spirited conversations, sometimes openly disagreeing with each other’s take on situations, but also demonstrating closeness and care.

Blake had been exploring his gender and sexuality for a few years. He reported feeling both like a boy and non-binary, as well as bisexual, but expressed no desire to change his name or pronouns from those assigned at birth. He described liking make-up, high heels, and dresses. At times, Faith questioned this and Blake would respond adamantly that liking these things did not make him a girl. He was allowed to start wearing make-up in public, including at school, when he started the 6th grade, the same as his older sister had been.

Blake was in both family and individual therapy. Faith shared that Blake started individual therapy at a local clinic for multiple reasons, of which gender and sexuality were a part, but that his therapists would only discuss gender and sexuality if he brought it up. Faith expressed concern that this was about their discomfort with TGD issues. Blake shrugged, indicating he had no concerns about his gender or sexuality. Blake had not told his primary doctor about his gender or sexuality, but Faith reported feeling as if they would be competent. Blake indicated no desire for any medical transition or gender-related care at this time. Faith shared that the biggest hurdle she witnessed regarding TGD-affirming care was in education for mental health providers. As a clinician herself, she shared that psychologists and social workers needed better and more ongoing continuing education on working with LGBTQ+ youth.

#### 3.1.4. Family 4

Ethan was four years old, living with his twin sister, mom, and dad in a small town. Situated in a more conservative and less densely populated area of the state, the town was unique in having a county-wide non-discrimination ordinance that included gender identity. There are no gender clinics or specific TGD-affirming providers in the community; however, there is a human rights organization providing training and resources to physicians providing care for all genders. Additionally, a more urban area, about an hour’s drive away, hosts additional resources for access to care. Ethan, his mom, Natalie, and their family were well-established in the community and the family had insurance through Natalie’s job. Ethan loved coloring, playing with toys, and laughing with his sister. During our visits, the family was loving toward each other and reported intentional efforts at building family cohesion and community.

Ethan began talking about his gender as a boy at 2 ½ years old. He regularly said he was a boy, would wear boy clothes, and questioned his own body. When we first met with Ethan and his family, he was using his name provided at birth and she/her pronouns. He appeared shy and reserved, though warmed up quickly and wanted to play with us. When we visited the second time, his demeanor had changed completely. He was using the name Ethan and he/him pronouns, had cut his hair, and was expressing his gender as a boy full-time. Natalie informed his preschool teacher, sharing that she “didn’t give them a choice” about whether to accept him or not. She said his Kindergarten the next year had a gender-inclusive policy, and she was not worried.

Natalie shared that Ethan’s general practitioner was open and supportive of Ethan and the family, noting no potential concerns. She shared that she sought out the expertise of a child therapist to identify how best to support Ethan, but the therapist shared harmful rhetoric surrounding a gender binary as opposed to being helpful. Although Ethan was too young for any TGD-specific medical care, Natalie shared concerns about being able to access that type of care in their town as Ethan got older. She indicated a barrier to effective care for Ethan is in the lack of education and training of medical and mental health providers.

These family descriptions illustrate how families navigated TGD youth healthcare, broadly and related to gender. Importantly, all the youth in the study had supportive parents willing to advocate for them to access affirming care. Most had the financial resources, insurance, and/or ability to travel to access care. This is clearly not the situation for many TGD youth; however, the themes that emerged across the families revealed important implications for access to TGD healthcare, likely relevant for TGD young people across contexts.

### 3.2. TGD Healthcare

Thematic analysis across data sources and families resulted in two themes related to healthcare: accessibility and affirming care.

#### 3.2.1. Accessibility

Youth and their parents described accessible healthcare as access to TGD healthcare information, having TGD-competent providers, and safe physical resources that were obtainable given their financial and geographical location. Each family shared stories about accessibility, recognizing their own privilege of access due to supportive parents with stable incomes and access to transportation. Even so, each family shared accessibility challenges including a lack of TGD healthcare information/providers, material hardship (finances, distance), and safe physical resources. Importantly, these challenges intersected to either enhance or further limit accessibility.

Participants discussed challenges in finding information related to local healthcare providers who could work with TGD youth. One parent said:


*We didn’t know where to go; doing random Google searches does not get you what you need, and looking that up on our providers care site, there was nothing. I mean, I could find 50 endocrinologists, but none of that told me if they would be able to help with this situation.*


While an internet search revealed resource lists of health providers in some communities in the state, many communities had no such information consolidated or readily available online. In addition to online searches, participants noted that they also sought information regarding TGD healthcare from their child’s primary doctor. They noted that, while well-intentioned, their child’s doctors often did not have referrals for healthcare services for TGD youth. Ultimately, most families in the study relied on information provided by other TGD youth or organizations that provide support for TGD youth and their families. A youth shared how they were able to access information word-of-mouth through other TGD connections: “if I didn’t go to the [LGBTQ] center, and if I didn’t meet my friend who told me about [TGD healthcare provider], I don’t know what would have happened.”

Even when families found and accessed TGD-affirming physicians, they shared concerns about high turnover rates and a general lack of physicians in their areas. This was particularly salient for rural TGD youth. One youth shared feeling as if they had to re-establish rapport regularly due to changing physicians at the clinic, speculating that turnover could be related to working in a high-stress clinic that also provides abortion services, where protesters regularly gather and individuals experience harassment when entering and exiting the clinic.

Safety, therefore, was of particular concern to the youth and parents. They described accessing gender-affirming care through “women’s clinics”, walking through protesters, and being yelled at. One youth said:


*Where I go, it’s [women’s clinic]. So they do more women’s health care. They do abortions and stuff there, too, so it’s like that’s why they have that much security cause you go through a metal detector, and the door’s locked. And then he has to unlock the door for you…But then they also do trans stuff on top of that.*


Another youth described the safety planning required to enter the facility where they accessed services.


*There are lots of anti-abortion people. They’re all like, there’s a building right next to them, and they have a sign that’s like, ‘Planning getting an abortion? We can help you, because we’re religious.’ They have a big truck with a really graphic image on it, which is probably fake, and they have tables and flyers. Before my first appointment, they called me and was like, ‘When you get there, just drive on in like you work there. Just ignore them, and they probably won’t try to talk to you.’*


Experiencing protesters, having to go through security protocols, and other similar experiences were common for youth accessing TGD healthcare in a women’s clinic setting.

Finally, families described material hardships related to accessibility such as affordability, insurance restrictions, and travel to access resources, even in homes with adequate family support and resources. Parents experienced insurance companies’ coding mistakes that led to lengthy conversations with Human Resources to ensure that hormones and other medical needs were covered for their child. They also described instances in which visits were not covered by their insurance provider, but they were unaware until after the appointment was billed and sometimes after multiple appointments. TGD youth were also aware of the costs for medical care. One participant said:


*I mean, I have a friend, and he’s getting top surgery in June or July, which is awesome for him. But I’m not. And it’s very expensive. I know, I have to wait at least another two years, if that, if I have the money. So it’s just kind of bittersweet, I guess. Because I’m really excited and happy for him. Because he’s been wanting this since- For so long.*


Travel was another accessibility issue. Most of the youth in this study had to travel, sometimes an hour or more, to access a physician who competently addressed their healthcare needs.

#### 3.2.2. Affirming Care

Affirming care was described by youth and their parents as affirming practices within the healthcare setting, having knowledge and attitudes that met their needs, and affirming spaces to access healthcare. Families identified specific instances in which name and pronoun use, providing education and referrals, and gendered space impacted their perceptions of the care they received as affirming or presented challenges to accessing affirming care.

Affirming practices were identified by participants as a broad range of affirming behaviors exhibited by or desired from healthcare providers, including using correct names and pronouns, requesting consent before engaging in services, and providing education to TGD youth as they engaged in hormone therapy. These behaviors significantly impacted participants’ perceptions of their experiences.


*I mean even just the normal things, asking pronouns. One thing I’ve been reading some about is consent and this isn’t just for trans (people), but obviously it could make a big difference. Even when the doctor comes to check your pulse, ‘Is it okay if I grab your arm and then I’m going to check.’ …Just that kind of consent would be in having a more trusting and comfortable relationship.*


One family in the study discussed an event that had a significant impact on their experience. A parent described it, saying,


*She brought us in, and she went over it all, she showed us everything. I mean, she went into detail. And then she gave us a sharps bucket, she gave us all kinds of extra needles for practicing, and for using, gave us all this stuff. It was like this, we call it his little trans starter kit, she gave us. She was so, I mean, that made such a world of difference for us. She was amazing…*


Affirming behaviors from medical professionals created an environment in which TGD youth felt more comfortable accessing the care they need.

Participants also relayed concerns about healthcare professionals’ education on gender-affirming healthcare and their inability to provide safe and effective referrals. Participants regularly went first to their general practitioners for references for gender-affirming care.

*We didn’t know where to start. We really like our general practitioner, so we just set up his regular physical and started by talking to her, and she said, ‘Well, I don’t feel comfortable with that. I’m not an endocrinologist, and I don’t know enough about it,’ and she said, ‘And I would not recommend the endocrinologist in town. There’s only one.’ And she said, ‘That’s just, he does not seem like the right fit.’ She tried to help us, but we ended up getting a reference from a friend at the Center, and we go to the* (Clinic’s name).

Participants noted that their practitioners seemed uncomfortable and uninformed when discussing gender-affirming care and saw gender-affirming referrals through their general practitioner as a barrier to finding care. “I think it would be easier to access that stuff through your family doctor, just going, and if you would tell them, and then they would be like, ‘Oh here’s a bunch of things we can do for you, whichever works best.’” One participant acknowledged the discomfort expressed by their general practitioner when a referral was requested, “I think doctors should be more educated on the topic because my doctor doesn’t really know a lot about trans people in general. She has a general idea, but she doesn’t know anything pretty much.” Many participants were able to find a referral to a practitioner who offered affirming care through interpersonal relationships, but they expressed concerns about accessing healthcare for other medical needs.


*Yeah, and I mean so far we’ve been very lucky. He’s got a great primary care physician who’s… We’re fine. But just, every time we see somebody new, I mean I made a point when choosing a primary care physician that that would be something on the list. But if he ever has to see a specialist, or if he ever has to go to the emergency room, who knows what we’re going to find?*


Concerns about accessing healthcare professionals outside of gender-based care, such as the emergency room or other emergent healthcare needs, was a concern for participants.

Additionally, participants acknowledged the need for those healthcare professionals who provide gender-affirming care to remember that youth accessing their services are not medical professionals themselves and require education and information about the care they are receiving. One participant said, “I think for young trans kids, I feel like they just…I didn’t know a lot even, and I feel my doctors didn’t really tell me enough. And I never really got what was going on. I would ask my mom and she would just tell me we’re moving in the right direction.” There may be a presumption from medical providers that youth accessing their services understand gender-affirming medical care because they sought out those services.

Gendered spaces also impacted participants’ access to affirming care. Specifically, a transmasculine youth shared discomfort about accessing women’s clinics.


*It kind of made me uncomfortable at first. It still kind of makes me uncomfortable. Because I was talking to my friend, and he’s cis. And I was just like, “Oh yeah, I go to (clinic’s name).” And it was like, uh, women… Not women, but it says women in the title. And it’s pink on the card, and just like, I’m not a woman, but I go there. So it makes me kind of uncomfortable.*


Another youth participant noted discomfort accessing services in a “women’s” space.


*I’m like, ‘Oh, I’m not a woman, but I’m going here,’ and if I’m walking, I was like, ‘I hope no one sees me walking in here.’ Or if I talked to other people, they’re like, ‘Oh, where do you go?’ I was like, ‘Oh, I go to (Clinic’s name).’ It was like, ‘What?’*


Participants mentioned a desire to have a space that was broadly available for gender-based care, “It would be amazing if there were a center here. Just gender in general, gender medical-based center that you could go to for all those things.” Even if women’s clinics are able to provide gender-affirming care, the gendered nature of the center can impact the affirming care TGD youth receive in those spaces, even if healthcare professionals there provide affirming care for TGD youth within the facility.

## 4. Discussion

This study explored the healthcare experiences of four TGD youth and their parents in one Midwestern state. Through extensive interviews, two broad themes arose that were explored in this paper: accessibility and affirming care. Findings reveal that even in supportive families with access to financial resources and community support, TGD youth face barriers to accessing affirming healthcare. Specifically, accessibility was limited by lack of information on TGD healthcare providers, financial resources and insurance restrictions, and safety of physical locations. Affirming care included knowledgeable and affirming providers and healthcare spaces and specific practices that demonstrated competence and care to TGD youth. These findings enhance our understanding of healthcare among TGD youth in the Midwest U.S.

A key finding from this study is the fact that even in supportive families, TGD youth experienced barriers to accessing TGD-affirming healthcare, including finding resources, financial barriers, and geographic limitations. Parents regularly discussed spending hours on the internet, talking to other parents, visiting LGBTQ+ resource centers, and calling doctors and clinics to find competent and affirming care for their child. This echoes findings from recent research that TGD youth lack access to competent healthcare providers [21,22,23]. Importantly, each family in this study had adequate health insurance coverage and yet still faced financial barriers to care, including travel to access resources and insurance challenges. Prior research has identified insurance and finances as a barrier to accessing TGD-affirming care [21,22]. Travel to access gender-affirming healthcare has not been specifically examined in TGD healthcare literature; however, it has been identified as a barrier to TGD and sexual minority youth access to affirming community-based resources [29].

Another finding from this study illustrates the importance of affirming practices from medical and mental healthcare providers. Several of the families in this study started their gender-affirming healthcare experience with the child’s primary care physician. Many of those providers were uncomfortable providing care or making referrals. Families also worried about what kind of care they might receive if their TGD youth had to go to the emergency department. This suggests that access to gender-affirming care is not just about access to gender-specific care, but also access to medical and mental healthcare providers and systems that are affirming regardless of what type of care a person needs.

### 4.1. Limitations

This study has important limitations to note. As an exploratory multiple case study, the findings should not be generalized to other populations or geographic settings. The nature of the study required parental permission and involvement, which meant parents who agreed to participate were more supportive than in other families with TGD youth. Finally, despite attempts to diversify the sample by both race/ethnicity and gender identity/expression, the final sample of families were all white and had TGD children who were transmasculine or non-binary; youth/families of color and transfeminine youth were not included. We make conclusions carefully considering these limitations.

### 4.2. Implications

The findings from this study are relevant to the current sociopolitical context for TGD youth and, combined with other evidence, have important implications for policy and practice. Given the rise of anti-TGD legislation and rhetoric, including particular emphasis on reducing access to healthcare services for TGD youth [9], it is critical that social workers, healthcare providers, and other professionals advocate for inclusive policies that promote access to affirming care for TGD people. Resisting anti-TGD legislation is an important place to start; however, professionals and advocates should also promote gender-inclusive policies at city, state, and national levels. Advocating for non-discrimination policies that are inclusive of gender identity and expression is an essential step toward promoting healthcare access and reducing discriminatory practices. Opposing additional access and affordability barriers to healthcare for TGD youth is both an ethical imperative and grounded in sound policy reasoning. The nondiscrimination provisions of the Affordable Care Act and other federal law, such as the Title IX civil rights provisions barring discrimination based on sex, may be interpreted as protecting the rights of TGD youth to equal access to care [30].

Additionally, this study’s findings highlight the enormous challenges which insurance policy and practice can pose for families of TGD youth. The families in this study all had access to private health insurance, a status associated with fewer insurance denials of TGD-related healthcare among adults, as compared to those with public health insurance [31]. Nonetheless, families reported routine insurance challenges, indicating the importance of improving both insurance policy and practices. From a policy perspective, the challenges families faced may reflect the important state and regional differences which motivated the present study’s attention to experiences in a Midwestern state. Current data on state-level TGD exclusion and inclusion in health insurance coverage indicate that nearly all Midwestern and Southern states do not have policies requiring coverage of TGD-related healthcare by private health insurance companies [32]. Moreover, state Medicaid policy in several Midwestern and Southern states explicitly excludes TGD healthcare coverage [32]. With many states’ policies to draw on as models, the clear policy implication is to expand bans on TGD exclusion from private and public health insurance coverage.

In addition to policy change, insurance-related practices emerged as a barrier to families, suggesting the importance of extending education and training to people who work with insurance claims. This might include healthcare organizations’ billing department employees and insurance companies’ customer service staff. While these are not the clinicians who often have direct contact with TGD youth, their work in coding, checking, and resolving insurance issues requires sufficient education and training to understand the experiences and healthcare needs of TGD youth and therefore interpret and process insurance claims correctly.

At the practice level, all participants in this study noted a lack of training for medical and mental health providers related to working with TGD people. As providers become more aware of, and more TGD youth access healthcare services for other healthcare needs, such as emergency room services, the need for training for all medical personnel will continue to expand. Eisenburg et al. [33] advocated for physical and healthcare provider training on a regular basis, and creating a system for tracking who has been trained and who is willing to provide support for TGD youth. This would help not just with training healthcare professionals but with providing resources for families to identify potential doctors or clinicians for their child. Training should consist of topics such as TGD-affirming language, debunking myths about TGD people and youth, and other medically relevant resources [33,34]

Relatedly, a critical practice implication of this research is a greater need for access to TGD-affirming resources. Physicians, mental health providers, and other groups such as LGBTQ+ community organizations should create and sustain a TGD-specific resource guide [33]. Such a guide can support both providers and families with finding affirming and accessible healthcare resources, reducing time spent seeking resources, helping providers make referrals to affirming providers, and providing autonomy to TGD youth who might not have familial support. This type of guide could be similar to resource guides present in large cities and urban areas, but at the county level. Additionally, the resource guide should be put into a pamphlet or brochure to ensure access for people with limited or no internet access.

Given our focus on youth, it is critical to discuss the importance of medical decision-making for TGD people under eighteen. Clark and Virani [34] explored the ethical concerns of providing medical care for TGD youth and found that TGD youth have the individual capacity to make decisions related to their own gender care. Importantly, the age of consent for medical treatment may vary by state and medical intervention, creating a potential barrier for access to affirming care. Even providers skilled in TGD care should expand their knowledge of frameworks such as positive youth development and how it intersects with TGD justice and empowerment [34].

Finally, there are important implications for research to note. We focused our study on one state in the Midwest; future research should continue to explore understudied geographic areas, including the Midwest, but also rural communities, the Southern U.S., and relevant global contexts. Additionally, future research should consider how to promote active and inclusive participation from participants while also generating findings that are more generalizable to larger groups of TGD youth. Survey methods that compare the experiences of TGD youth across states with varying policy contexts and sociopolitical environments may also be useful. Finally, innovative and less oft used methods, such as multiple case studies, provide researchers with tools to explore healthcare and health experiences with TGD and other marginalized youth in unique and nuanced ways. It may be helpful to use this type of method with states or healthcare systems as the ‘cases’ to better understand TGD youth’s experiences within these different contexts.

## 5. Conclusions

Numerous studies have shown that TGD youth face multiple barriers to accessing gender-affirming care. This study explored those access-oriented and gender-affirming-care-oriented barriers for white TGD youth in the Midwest who have supportive families. Even with supportive and financially stable families, these families identified barriers to accessing gender-affirming care. These barriers included a lack of access to physicians, accessing gender-affirming care financially and geographically, a lack of information and resources about gender-affirming care, and safety concerns of accessing gender-affirming care. Families identified a lack of affirming care as a barrier to access when attempting to access information on healthcare providers with knowledge about or referrals for gender-affirming care for TGD youth, when seeking information or referrals from healthcare providers, and when required to engage gendered space to access gender-affirming care. These barriers present important implications for practice and further research.

## Figures and Tables

**Table 1 ijerph-18-09219-t001:** Family Case Characteristics.

Pseudonyms	Child Age	Child Gender	Child Pronouns	Child Race	Town/Region	TGD Healthcare Utilized
Carrie (mom) Travis (kid)	16	Boy	he/him	White	Rural/South	therapy; hormones
Vanessa (mom) Aiden (kid)	15	Boy	he/him	White	Suburban/North	therapy; hormone blockers; hormones
Faith (mom) Blake (kid)	11	Non-binary	he/him	White	Suburban/South	therapy
Natalie (mom) Ethan (kid)	4	Boy	he/him	White	Small town/West	therapy

Note. Pseudonyms are used to protect family anonymity.

## Data Availability

Data are not made available to protect the privacy of participants.

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
