# Peer review of "“I Have Nowhere to Go”: A Multiple-Case Study of Transgender and Gender Diverse Youth, Their Families, and Healthcare Experiences"

_ijerph, 2021, doi:10.3390/ijerph18179219_

Round 1
Reviewer 1 Report
This is a well-written manuscript that provides a starting point for continued research on the health care needs of transgender and nonbinary youth in the US Midwestern and Southern states. Although the generalizability of the study findings is limited by the sample size and some lack of diversity in the sample, the approach is novel and the authors present useful findings.
I have two minor suggestions for the authors:
- I believe it would be beneficial to add a short paragraph that focuses on research implications derived from your findings. You've started this important inquiry in one Midwestern state - what should you and other researchers go next from this?
-
Pg. 7 line 320 - “population” should be “populated”
I enthusiastically recommend this paper for publication following these minor revisions.
Author Response
Thank you for your kind words about our manuscript. We appreciate your review and suggestions to improve the manuscript.
- We added a paragraph to the end of the Implications section on suggestions for future research.
- We fixed this typo. Thank you for finding it.
Reviewer 2 Report
Very well done study and manuscript on an important topic. The comments below are minor suggestions, however, this manuscript is already quite strong as is.
I would like to see more detail in the methods section (in particular, pertaining to the data analysis).
In addition, while the results provided a good overview of each individual case, I felt the true analysis (interpretation of data) came in when the cases were compared to one another (though the manuscript indicated some level of analysis on individual cases).
Additionally, in several places, insurance/finances are mentioned. In particular, on p12, line 565, you mention that though privately insured, the families still discussed challenges related to insurance claims. However, these are never fully discussed - it would benefit the manuscript to do so.
Author Response
Thank you to the reviewer for your kind words and your thorough review of our manuscript.
- We added more detail to the data analysis section.
- Thanks for the important note about adding more about insurance challenges. We added some more detail to a couple of the family descriptions. We also address insurance across themes in the findings related to accessiblity.